# Development and optimization of red spectrum splitting concentrated agrivoltaic system for energy generation and sustainable agriculture

Ngoc-Hai Vu[1]*, Tran Quoc Tien[2]*, Ngoc-Minh Kieu[3], Thanh-Phuong Nguyen[4], Quang Cong Tong[2], Seoyong Shin[3]

1 Faculty of Electrical and Electronics Engineering, Phenikaa University, Hanoi, Vietnam, 2 Institute of Materials Science, Vietnam Academy of Science and Technology, Hanoi, Vietnam, 3 Department of Information and Communication Engineering, College of ICT Convergence, Myongji University, Yongin-si, Gyeonggi-do, Korea, 4 Faculty of Engineering Physics, Hanoi University of Science and Technology, Hanoi, Vietnam

* tientq@ims.vast.ac.vn (TQT); hai.vungoc@phenikaa-uni.edu.vn (NHV)

## Abstract

Agricultural land is increasingly under pressure from expanding solar energy infrastructure, leading to a growing conflict between food and energy production. Conventional photovoltaic (PV) systems often reduce crop yields due to shading and suboptimal light conditions for photosynthesis. This study introduces a Red Spectrum Splitting Concentrated Agrivoltaic (RSSCA) system designed to overcome these limitations by selectively directing specific wavelengths of sunlight to serve both agricultural and energy needs. The system uses a Fresnel lens and dichroic mirror to concentrate solar radiation and split the spectrum: red light (around 630 nm), essential for rice growth, is transmitted to the crops, while the remaining wavelengths are directed to high-efficiency multi-junction solar cells. We employed optical simulations and analytical modeling to assess system performance. Results show a photoelectric conversion efficiency of 31.2% and adequate Daily Light Integral (DLI) levels for rice cultivation in high-sunlight regions such as Hanoi and Ho Chi Minh City. For instance, DLI values in Hanoi range from 14.3 to 32.2 mol/m²/day and in Ho Chi Minh City from 24.3 to 37.8 mol/m²/day, compared to 6.6–16.1 and 14.4–18.2 mol/m²/day under conventional PV systems, respectively. Electricity production with RSSCA peaks at 2790 W/m²/day in Ho Chi Minh City and 2577 W/m²/day in Hanoi, outperforming traditional PV setups by 3–5 times. Compared to conventional PV systems, the RSSCA offers improved light uniformity, better land-use efficiency, and significantly higher electricity generation. However, in regions with limited direct sunlight, such as Seoul in winter, system performance decreases. These findings suggest the RSSCA system is well-suited for tropical and subtropical areas where direct sunlight is abundant. It presents a viable solution for enhancing both food production and renewable energy generation on shared land.

**Data availability statement:** All relevant data are within the paper.

**Funding:** This research was supported by the Vietnam Academy of Science and Technology (Grant Number QTKR01.03/23-24). There was no additional external funding received for this study.

**Competing interests:** The authors have declared that no competing interests exist.

## Introduction

As renewable energy becomes a global priority, solar and wind power are key drivers in the transition away from fossil fuels. By 2024, solar capacity is expected to reach 1.6 TW, aiding carbon reduction efforts. However, large photovoltaic (PV) installations compete with agriculture for land and raise waste management concerns, thereby necessitating further research and policy intervention [1].

Agrivoltaic systems are rapidly advancing as a dual-use solution, combining solar energy production with agricultural growth, and are increasingly being implemented worldwide to address land use challenges and enhance sustainable farming practices. This approach optimizes land use by integrating solar panels with crop cultivation, livestock grazing, or other agricultural activities, providing mutual benefits for energy production and farming. Despite current limitations related to initial investment costs and the need for research and development tailored to specific crops and regions, agrivoltaics has significant advantages in enhancing land-use efficiency. With its potential to contribute to both sustainable energy production and agricultural productivity, agrivoltaics is poised to become a prominent global trend in the near future [2].

Current projects and research on agrivoltaic systems demonstrate their growing adoption and adaptation worldwide, with a variety of innovative examples illustrating their potential to integrate solar energy production with agricultural activities. Agrivoltaics system studied by Dupraz et al. (2011) [3] has shown promising results by combining solar panel installations with crop cultivation, enabling both energy generation and agricultural productivity on the same land. Similarly, in the United Kingdom, farms have successfully adopted agrivoltaic practices, integrating solar panels with traditional farming methods, as reviewed by Ravi et al. (2020) [4]. This dual-use approach has gained traction as a way to maximize land use and enhance both energy and food production. In Japan, particularly in rural areas, the implementation of agrivoltaic systems has been coupled with comprehensive technical, economic, and institutional analyses, providing valuable insights into their potential benefits and challenges (Kato et al., 2020) [5]. Typically, these systems utilize the spaces beneath solar panels for growing crops, relying primarily on the light that filters through the gaps between the panels. However, to achieve high agricultural yields in agrivoltaic systems, the spacing between solar panels must be sufficient to allow adequate light for crop growth and development. While this setup benefits the crops, it can also result in a considerable reduction in electricity output, as the sunlight is less concentrated on the panels. Consequently, the energy production per unit area is lower compared to traditional PV systems [6]. To overcome this challenge and improve solar power generation efficiency, Concentrated Photovoltaic (CPV) technology is being explored as a promising alternative. CPV technology offers the advantage of focusing sunlight onto small, high-efficiency photovoltaic cells, achieving a higher energy conversion rate than conventional PV systems, while also supporting crop growth [7].

CPV systems integrated with agricultural cultivation are being studied worldwide to enhance the overall efficiency of a unit area. Sato et al. proposed two highly transparent CPV module models for efficient dual land use, achieving an electrical efficiency

of 26.7%. Type A modules, using a typical-scale CPV lens, fully separate direct-diffuse light for energy applications, while Type B modules, with micro-scale CPV lenses, enhance direct sunlight transmission by reducing the lens-to-module aperture ratio. Both models demonstrated higher power output and transmittance compared to standard semi-transparent PV modules with 17% efficient Si cells. [7]. A significant scientific advancement here is the development of CPV modules that allow a substantial amount of light to pass through to the crops below while still capturing part of that light for electricity generation. Daiki Hirai et al. developed an experimental CPV module with a photoelectric conversion efficiency of 28% and a transmittance of 70% with diffused light [8]. The 70% transmittance with diffused light means that the CPV modules not only concentrate light for electricity generation but also transmitted the diffuse light, which is beneficial for crop growth. Harry Apostoleris et al. proposed a model of high-concentration photovoltaics for dual-use with agriculture, with CPV output assuming 30% power conversion of the direct solar component [9]. These studies show improved land-use efficiency, but in agrivoltaic systems, crops primarily receive diffuse light, as direct sunlight is absorbed by the panels. While diffuse light aids growth, it often lacks the intensity and specific blue (470 nm) and red (630 nm) wavelengths crucial for photosynthesis, especially for tropical crops. Optimizing both direct and diffuse light is essential to support sustainable crop growth alongside electricity generation.

Recent studies have introduced more sophisticated agrivoltaic designs that incorporate dynamic spectral management, smart materials, and integrated climate control to enhance dual land-use performance. Dinesh et al. (2022) [10] investigated the use of tunable optical filters to dynamically adjust transmitted light spectra, improving crop yield and energy output under varying weather conditions. Similarly, Jiang et al. (2023) [11] proposed an electrochromic-based agrivoltaic system capable of real-time spectrum control tailored to crop growth stages. In another recent development, Park et al. (2023) [12] implemented a hybrid agrivoltaic greenhouse integrating CPV modules with spectrum-selective coatings to optimize productivity in subtropical regions. Despite these advances, most recent studies have focused on shade-tolerant vegetables or greenhouses, with limited emphasis on open-field applications for high-light-demand crops such as rice. Moreover, there remains a notable gap in the development of CPV-based systems with spectrum-splitting optics explicitly designed to match plant-specific physiological requirements. Addressing this gap, our study introduces a novel Red Spectrum Splitting Concentrated Agrivoltaic (RSSCA) system tailored to rice paddies. This system uniquely combines red-light targeting with CPV efficiency, presenting a feasible solution for sustainable food–energy co-production in high irradiance tropical environments.

Rice is the most widely cultivated staple crop in Asian countries, and the abundance of flat land offers an advantage for setting up solar power systems. However, integrating rice cultivation with solar energy generation remains challenging due to rice's high sunlight requirements, which range from 15–25 mol/m$^2$/day depending on the growth stage [13–15]. Rice particularly requires red light (630 nm) and blue light (470 nm) [16]; blue light has a greater impact on morphological development and leaf growth, while red light is crucial for promoting flowering and influencing the ripening of fruits and seeds. Agricultural research suggests that a red-to-blue light ratio of 3:1 provides the greatest benefit for rice growth and development [17]. For rice cultivation, the systems mentioned above may be unsuitable, as most rely primarily on diffuse light, which is better suited for shade-tolerant crops.

To address the aforementioned challenges, we propose a Red Spectrum Splitting Concentrated Agrivoltaic (RSSCA) system. The RSSCA system is designed to be applied in rice paddies, offering enhanced electricity compared to conventional PV systems while ensuring the light requirements necessary for optimal growth and development of rice plants are met. This approach considers critical factors such as light spectrum needs, light intensity, and light uniformity within the cultivation area-elements that are frequently overlooked in existing CPV systems. Fig 1 presents the block diagram of the proposed RSSCA system. In this system, concentrated solar panels are designed to allow sufficient light penetration for the rice plants to photosynthesize effectively. This optimization of resource usage enhances productivity, moreover, the system minimizes environmental impacts and boosts the self-sufficiency of rural communities, contributing to sustainable development and resilience to climate change.

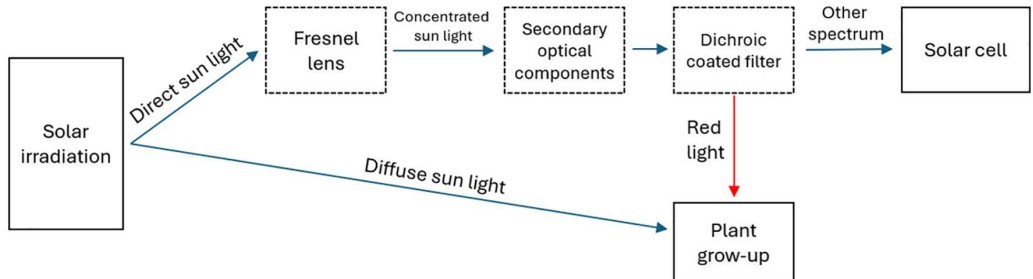

**Fig 1. Block diagram of the RSSCA system integrated for energy harvest and agriculture.**

Despite the progress in agrivoltaic technologies, existing CPV systems are not specifically optimized for high-light-demanding crops such as rice, particularly in terms of spectral and spatial light distribution. Most current systems provide diffuse or filtered light, which is insufficient for crops requiring high-intensity direct red and blue light essential for photosynthesis and crop development. Furthermore, conventional CPV designs rarely address the nuanced needs of tropical agriculture, especially in regions with intense solar radiation but limited land availability.

This study introduces a novel Red Spectrum Splitting Concentrated Agrivoltaic (RSSCA) system that bridges this critical gap. Unlike previous systems, the RSSCA is uniquely engineered to simultaneously maximize photovoltaic efficiency and deliver targeted spectral light specifically red wavelengths (around 630 nm) to support the growth of rice, a staple crop in Asia with high light requirements. The system's design, which incorporates a Fresnel lens, dichroic mirror, and solar tracking, enables efficient spectrum splitting and land use, addressing both energy and food security.

The significance of this research lies in its tailored approach for tropical, rice-growing regions, where balancing energy generation and crop productivity is a pressing challenge. By ensuring optimal Daily Light Integral (DLI) levels for rice while achieving a high photoelectric conversion efficiency of 31.2%, this work sets a new benchmark for sustainable agrivoltaic solutions. It offers a scalable model that aligns with global efforts to decarbonize energy systems and improve agricultural resilience in the face of climate change.

### RSSCA concept

This section describes the operating principles and optical design of the RSSCA system. Our proposed design separates sunlight into two spectral components to serve dual purposes: electricity generation and agricultural use. As shown in Fig 2, a Fresnel lens concentrates sunlight onto a secondary optical component – a prism coated with a dichroic mirror – which allows red light (630 nm) to pass through to the agricultural area. Light of other wavelengths is redirected to high-efficiency photovoltaic cells for power generation. These components are mounted on a solar tracker to ensure precise sunlight alignment throughout the day.

Red light in the 630–750 nm range makes up only about 15–20% of the visible spectrum's power, or roughly 8% of total solar radiation [18]. Because diffuse light arrives from all directions, it is difficult to focus and utilize effectively for electricity generation in this system. Since rice requires an optimal red-to-green light ratio of 3:1 for healthy growth, a red-light filter is used [19]. In this setup, sunlight concentrated by the Fresnel lens is directed to a secondary optical component, where it is split, red wavelengths are transmitted to the rice fields to support growth, while the remaining spectrum is redirected to high-efficiency solar cells for power generation. However, this system's reliance on diffuse light more suitable for shade-tolerant crops may not fully meet rice's need for direct sunlight. As in previous Agrivoltaic systems, direct light supports electricity generation and diffuse light aids plant growth, yet the red-light component alone comprises only a small fraction of solar energy.

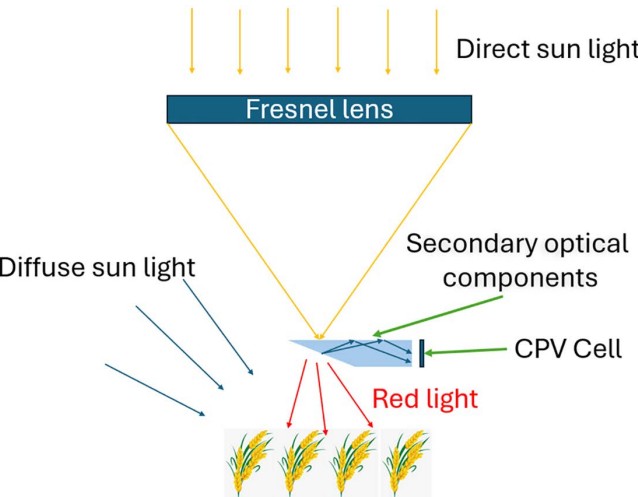

**Fig 2. Illustration of key system components: Fresnel lens, filter and high-efficiency solar cells.**

## Primary optics

Fresnel lenses as primary optics have become an important optical component in CPV systems thanks to their ability to focus light efficiently and at a cheaper cost than conventional focusing lenses. In our proposal, a commercial Fresnel lens size 250 × 250 mm and focal length 300 mm was used for the purpose of focusing sunlight on the secondary optics. The use of Fresnel lenses in this proposal in addition to the advantages of performance and low cost, the mass of the whole system is also reduced compared to the use of conventional focusing lenses, as the mass of the system is reduced, the pressure on the tracking equipment is also reduced, as a result, additional costs can be saved on the control motor.

We utilized LightTools [20] software to analyze convergence properties and evaluate the performance of Fresnel lenses. The input light source is a solar spectrum ranging from 400 nm to 2200 nm. The transmission efficiency of the Fresnel lens achieves 93%, and the concentrated beam of 20 mm. This determination of beam diameter is crucial for designing the system's secondary optical components. Fig 3 below illustrates the ray tracing (Fig 3a) and light distribution at the focal point (Fig 3b), simulated using LightTools software. In addition to determining the diameter and position of the converging beam, our simulation plays a vital role in assessing the overall performance of the RSSCA system.

## Secondary optics

The Fresnel lens focuses sunlight directly onto the prism (the secondary optical component) at the focal point. This optical component consists of a rectangular polymethyl methacrylate (PMMA) block featuring a partially beveled cross-section, with a dichroic mirror coating applied on the beveled surface. This coating reflects all wavelengths except the 630 nm region, redirecting the radiation toward the solar panel array. The red light in the 630 nm region passes through the filter and continues toward the agricultural area. Fig 4 illustrates the details of the secondary optics including waveguide, prism and dichromatic mirror to achieve high uniformity in sunlight distribution (a) and the light distribution at the location of the solar panel (b).

To ensure that the prism captures the entire light beam concentrated by the Fresnel lens and efficiently directs it to the solar cell, it must be designed to achieve total internal reflection. Based on beam size and Snell's law, we designed a 20 mm-wide prism to capture the full concentrated beam. Through precise calculations, we determined a 28° tilt angle for the

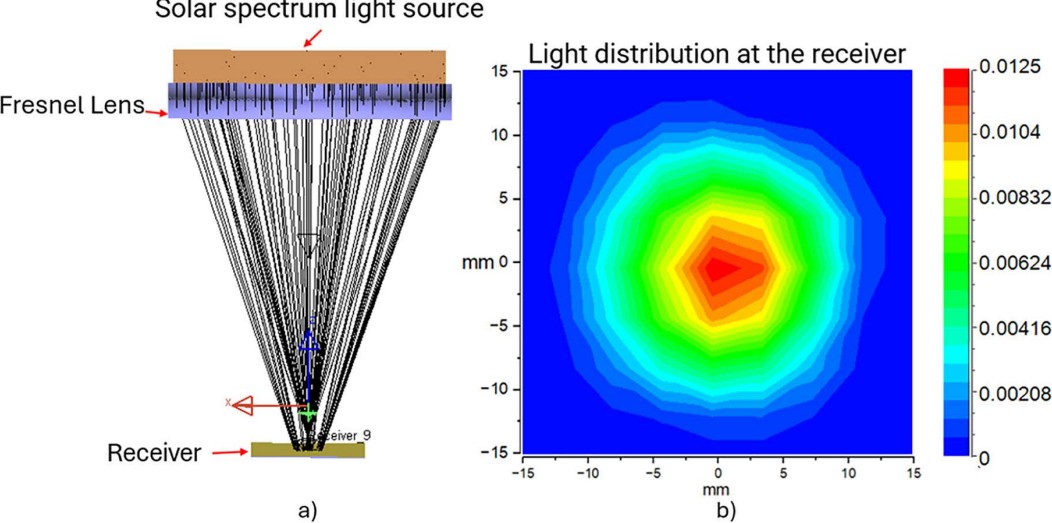

**Fig 3. Simulation results of sunlight transmission through the Fresnel lens and distribution at the focal plane receiver.** (a) Ray tracing of sunlight through a Fresnel lens and (b) sunlight distribution at the simulated using LightTools software.

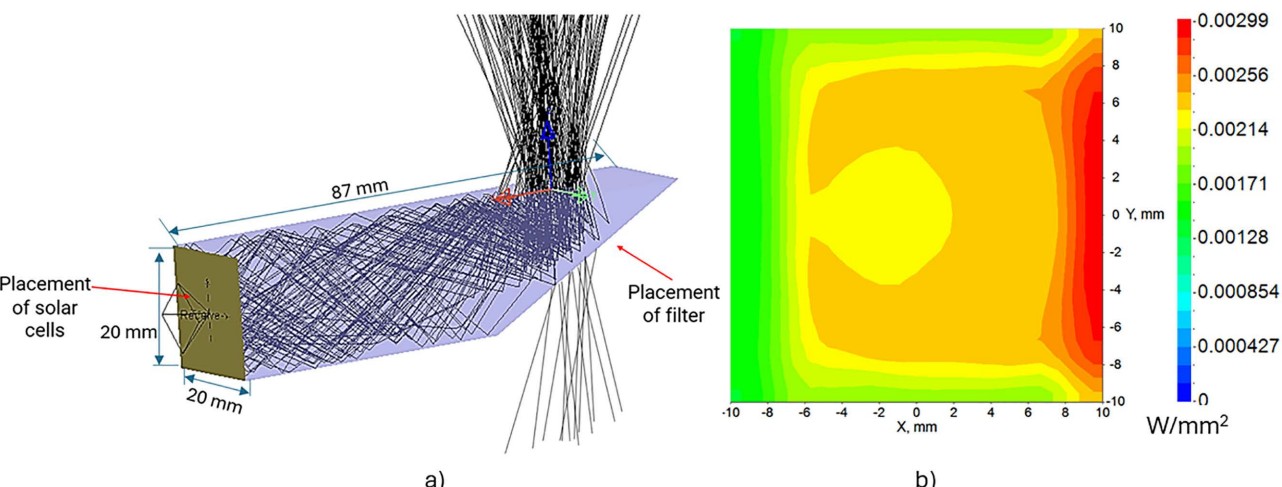

**Fig 4. Simulation results of optical design and sunlight distribution.** (a) Design of secondary optics, and (b) Sunlight distribution at solar cell placement.

prism to guarantee total internal reflection and achieve an optical efficiency of 84%. Moreover, the prism also serves as a homogenizer, evenly distributing sunlight across the solar cell surface with a uniformity of 70% on area of 40 mm².

After determining the optimal size for the prism, we proceeded to design a dichroic mirror to cover it, allowing the red-light beam to pass through to the cultivation area below. In this study, the dichroic mirror was configured as a Fabry-Perot resonator (FPR) [21], illustrated in "Fig 5".

In optical bandpass filter design, two dielectric materials are used to create multilayer mirrors and resonators, consisting of alternating high and low refractive index layers, each typically a quarter of the wavelength in thickness. This structure defines the stopband and passband characteristics. Fine-tuning the filter's performance involves optimizing mirror

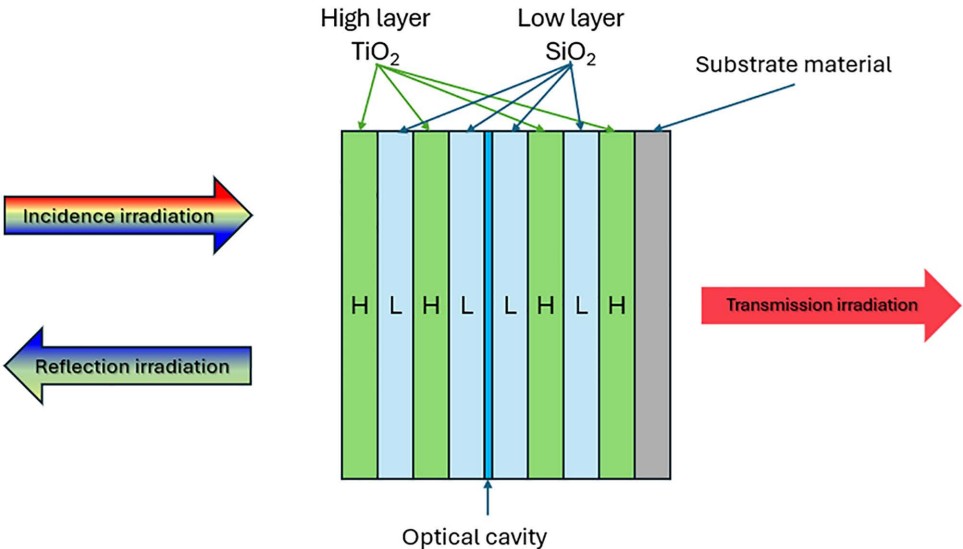

**Fig 5. The dichromatic mirror was designed as a Fabry-Perot resonator structure based on TiO$_2$/SiO$_2$ materials.**

reflectivity, using non-quarter-wave layers or multilateral mirrors. These techniques control passband ripple and bandwidth to meet specifications [22]. Our design applies the Narrowband Transmission Filter (NTF) principle for dichroic mirrors with a multilayer (HL)$^N$ structure at wavelength $\lambda_0$, where L and H are low and high refractive index layers, respectively, with equal optical thickness. In this configure ration, the optical thickness of the low index and high index layers is set to a quarter of the wavelength ($n_L L_L = n_H L_H = \lambda_0/4$). Were, $n_L$. and $n_H$ are the refractive index of the low index layer and the high index layer, respectively; $L_L$ and $L_H$ are the thickness of the low index layer and the high index layer, respectively. Consequently, the (HL) structure functions as a dielectric mirror at wavelength $\lambda_0$. When $\lambda_1$ and $\lambda_2$ are the left and right band edges in $\lambda_0$, the reflection coefficient $\rho$ can be defined as follows [23]:

$$\rho = \frac{n_H - n_L}{n_H + n_L}$$

(1)

The solutions for the left and right band edges and the bandwidth in $\lambda_0$ are:

$$\lambda_1 = \frac{\pi(n_H L_H + n_L L_L)}{\text{acos}(-\rho)}; \ \lambda_2 = \frac{\pi(n_H L_H + n_L L_L)}{\text{acos}(\rho)}; \ \Delta\lambda = \lambda_2 - \lambda_1$$

(2)

The dichromatic mirror structure is designed by copying (HL)$^N$ into (HL)$^N$(HL)$^N$ then a quarter of the wavelength L is inserted between the two groups, resulting in (HL)$^N$ L(HL)$^N$ being an FPR structure that opens the wavelength propagated at $\lambda_0$ between the reflector $\Delta\lambda$. In this study, dichromatic mirrors applied formulas (1) and (2) to calculate TiO$_2$/SiO$_2$ (HL) coatings with FPR for wavelength 630 nm, the refractive index of the substrates and the L and H layers were: $n_S = 1.52$(Glass), $n_L = 1.4$(SiO$_2$), and $n_H = 2.1$(TiO$_2$) and gave the result $\Delta\lambda = 462$ nm. Following design and simulation in LightTools software, we developed a 50-layer TiO$_2$/SiO$_2$ dichromatic mirror. Fig 6 presents the simulation results for the filter across the 400–700 nm spectral range. The transmission efficiency at peak 630 nm is 90%.

However, these results were obtained through simulations using a parallel incident light beam. To further understand the system's performance, we continued by simulating the combination of the designed filter with additional optical

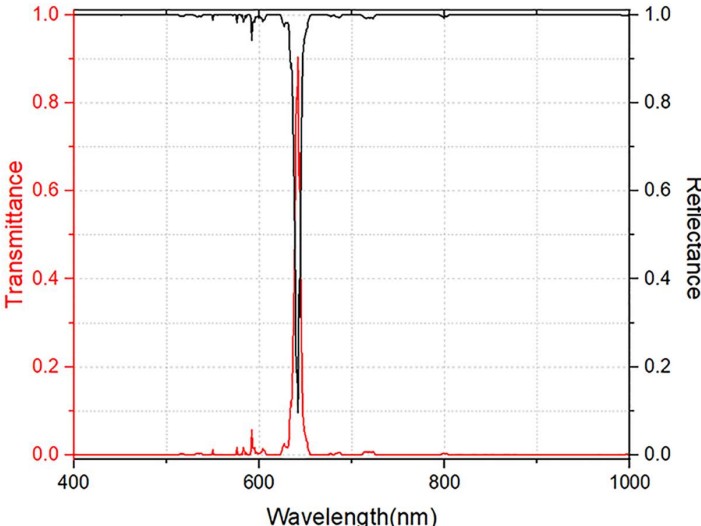

**Fig 6. Simulation results of dichromatic mirror across the 400 - 700 nm spectral range.**

components. Fig 7 shows the transmission spectrum when compared to the solar spectrum in the 400–1000 nm range. Fresnel lens and filter-coated prism. The transmission spectrum width is broadened compared to Fig 6, as the prism is designed with a beveled cross-section of 28 °.

   When the optical components, including the Fresnel lens and the filter-coated prism, were combined, the transmission efficiency at the 630 nm peak dropped to 85%. The filter allows light to pass through in the wavelength range from 582 nm to 695 nm, with a spectral bandwidth of 90 nm. This reduction in efficiency and the broadening of the transmitted spectrum can be explained by the influence of the tilt angle of the prism, combined with the incident angle of the concentrated beam. These factors cause a spectral shift in the transmitted light, which had been previously analyzed in our earlier

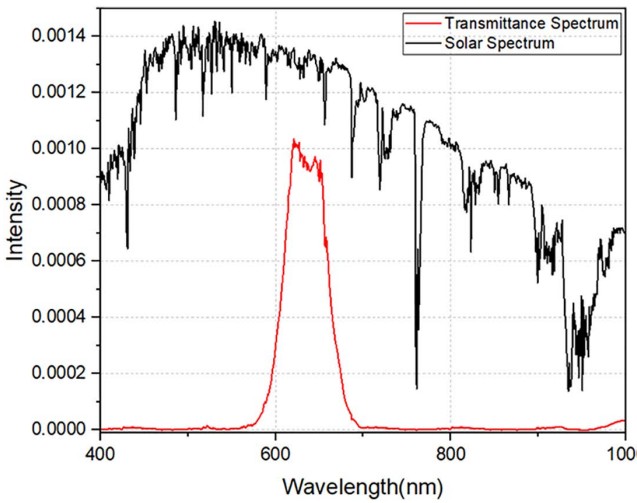

**Fig 7. Transmittance spectrum after combining optical components.**

simulations. The combination of optical elements affects the filter's performance, and understanding these interactions is key to optimizing the overall system.

After completing the calculation and simulation of the system's optical components, we proceeded to evaluate the overall performance and efficiency of the system. Fig 8 provides an overview of the primary components considered in the simulation.

The system's losses can be attributed to some contribution: absorption by the optical elements, Fresnel lens losses and beam leakage, both of which play a significant role in reducing the system's overall efficiency. For the Fresnel lens, the losses were primarily due to Fresnel reflections at the surface, as well as absorption within the lens material itself. These combined losses were calculated to account for 7% of the total energy. Once the light passed through the prism, the portion of red light successfully transmitted represented 8% of the total energy, while 78% of the energy was retained through internal reflection within the prism. Additional losses, such as beam leakage and absorption within the prism material, contributed another 7% to the overall loss. By factoring in these values, along with the efficiency of the multi-junction solar cell used in our design is 40%, we calculated that the total photoelectric conversion efficiency of the RSSCA system reached 31.2%. This result highlights the impact of both optical design and material properties on the system's efficiency and underscores the importance of minimizing losses in future iterations.

## System performance & discussion

**Overall system design.** In this study, a CPV system integrated with agriculture utilizing spectrum splitting technology was designed and prototyped. It includes the following main components: base, electric motor, frame, sun tracking sensor, Fresnel lens, shelf, filter, and waveguide. This system not only optimizes solar energy collection but also utilizes the split light to support plant growth, thereby enhancing both photovoltaic efficiency and agricultural productivity. By using spectrum splitting technology, light is divided into distinct wavelength bands, allowing each element of the system to operate more efficiently. The base and frame provide a solid structure, ensuring the system remains stable in all weather conditions. Electric motors adjust the orientation of the Fresnel lenses according to the sun's position, thereby maximizing

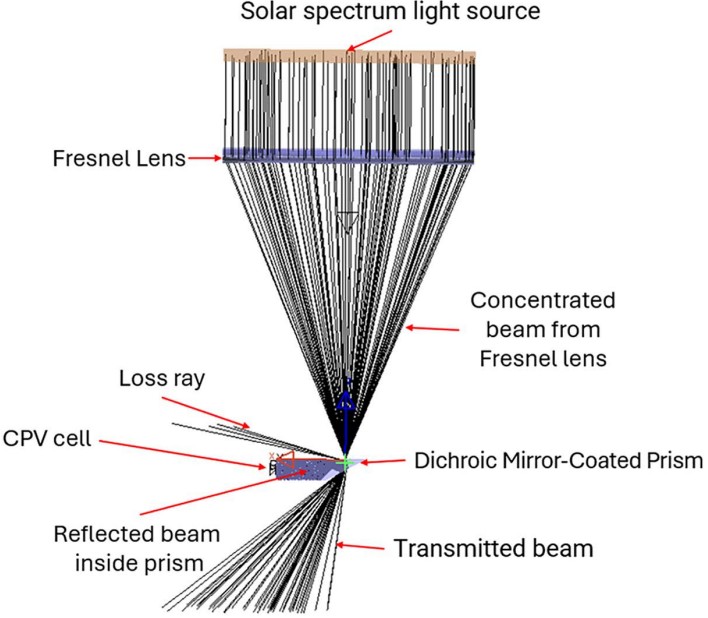

**Fig 8. Ray tracing of optical components in the RSSCA system: design and visualization of light paths.**

sunlight capture throughout the day. The sun tracking sensor plays a crucial role by determining the sun's exact position and controlling the electric motor to adjust the Fresnel lens angle. This optimizes sunlight concentration on specific points within the system, increasing collected light and improving the efficiency of converting optical energy into electricity. The shelf plays a vital role in adjusting the position of the waveguide block to align with the optimal focal point of the Fresnel lenses. This ensures effective delivery of the spectrum-separated light to designated locations, including photovoltaic cells and plant growing areas. Filters separate distinct wavelength bands, optimizing light use for both electricity generation and plant growth. This system not only maximizes sunlight utilization but also improves agricultural conditions. It creates a sustainable and effective model that combines energy production and agriculture. Fig 9 depicts the overall system design, clearly illustrating the components and their interactions within this integrated agricultural CPV system: base, electric motor, frame, sensors, Fresnel lens, shelf, filter, and secondary optics.

Following the completion of the RSSCA system design, a prototype was fabricated, and preliminary measurements were conducted to evaluate its performance. Fig 10 provides a comprehensive view of the system and its manufactured modules.

Specifically, Fig 10a illustrates the RSSCA system installed on a sun-tracking platform, designed to optimize the capture of sunlight throughout the day. Fig 10b offers a closer look at the secondary optics prior to their integration into the system, highlighting their design and construction. Finally, Fig 10c depicts the secondary optics in active operation within the RSSCA system, demonstrating their role in directing and optimizing light distribution for dual-use applications.

During the operation of the RSSCA system, we conducted detailed measurements and analysis of the light spectrum behind the filter to evaluate its performance. Fig 11 illustrates the transmitted light spectrum, clearly showing that wavelengths within the range of 610–645 nm are effectively transmitted through the filter, while other spectral regions are reflected. This behavior aligns precisely with the system's initial design parameters and its intended objectives. The targeted transmission of red light in this specific range ensures optimal light delivery for agricultural purposes, while the reflected wavelengths are directed toward the solar cells for electricity generation. These results confirm the effectiveness of the RSSCA system in achieving its dual-purpose design objective, enhancing both agricultural productivity and efficient solar power generation.

## Performance of RSSCA system

To evaluate the effectiveness of the proposed system, we conducted simulations and compared various parameters between our proposed system and a conventional PV system with the same land-use ratio. The compared parameters include light distribution in the crop-growing area, daily light integral (DLI), power generation capacity, and the efficiency of both systems. DLI is a key metric in the fields of photosynthesis and plant biology, measuring the total amount

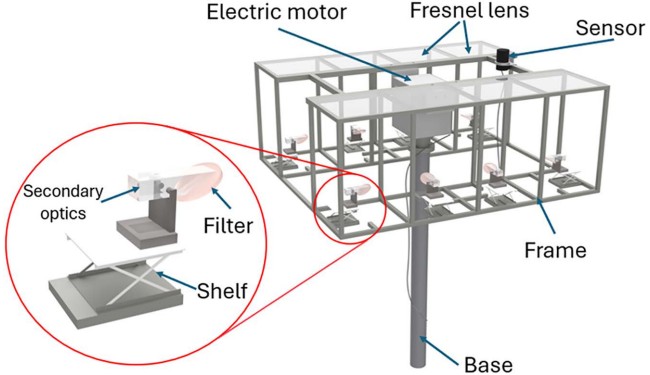

**Fig 9. Design of a RSSCA system for agriculture.**

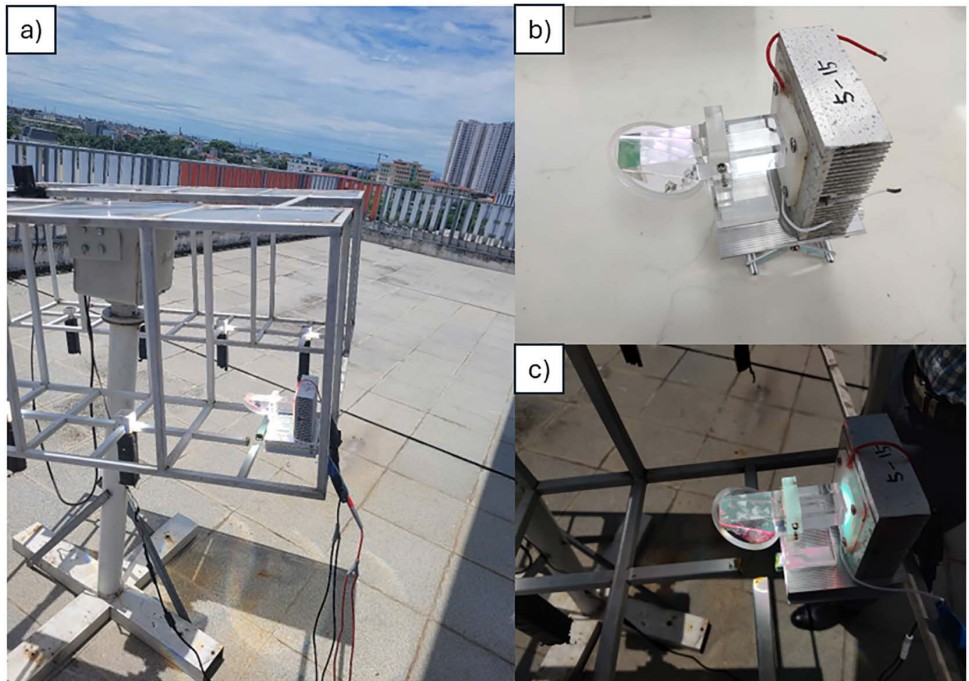

**Fig 10. Description of RSSCA prototype.** (a) Overall prototype system, and (b) Secondary optic after fabrication, and (c) Convergence of sunlight at secondary optics.

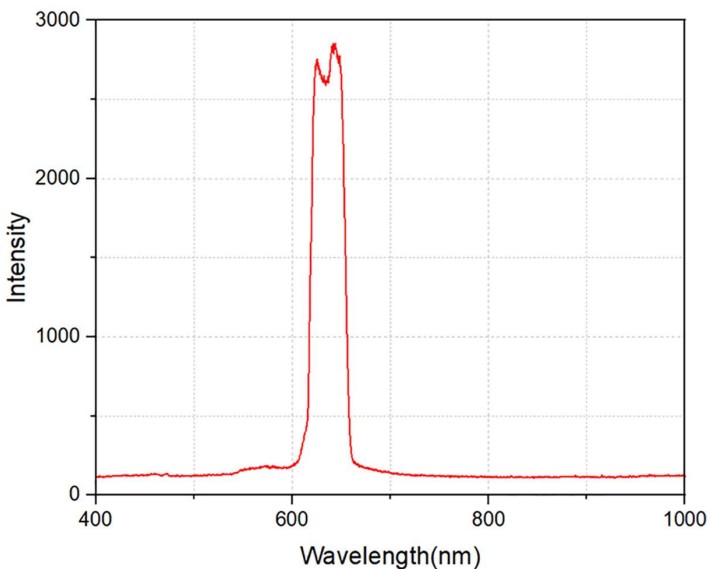

**Fig 11. Spectral transmission of the dichroic filter – selective red light filtering for optimized plant growth.** The filter transmits red light within the 610–645 nm range, matching the key absorption band of chlorophyll-a and supporting rice photosynthesis and flowering, while reflecting non-essential wavelengths toward solar cells for electricity generation.

of photosynthetically active radiation (PAR) that plants receive over the course of a day. PAR refers to light within the 400–700 nm wavelength range, which is utilized by plants for photosynthesis. The detailed configuration and comparative parameters are presented in the following sections.

RSSCA system: In our comparative simulation, we designed the total area of Fresnel lens panels to be 64 m$^2$ within a 100 m$^2$ plot. Each 1 m$^2$ section comprises 16 Fresnel lens panels measuring 250 mm x 250 mm. The system stands 3 meters tall, and the distance between the frame columns of adjacent systems is 1.5 meters. This design not only optimizes solar energy collection but also creates ample space beneath the panels for rice plants to grow and thrive. It also allows for the effective use of small and medium-sized agricultural machinery, harmoniously combining high technology with traditional agricultural practices.

Conventional PV system: The PV system utilizes rows of solar cells positioned parallel to each other, tilted 30° southward. This orientation maximizes energy harvesting efficiency, as locations in the Northern Hemisphere receive the most direct sunlight from the south. Each PV system also stands 3 meters tall, with a spacing of 3 m between the columns of each row. Additionally, a spacing of 2 m is maintained between the frames within the same row. This design not only optimizes solar energy capture but also creates an open structure that simplifies system maintenance and inspection.

By comparing these two systems, we can evaluate the effectiveness and feasibility of each design for capturing and utilizing solar energy in agriculture. "Table 1" provides a detailed breakdown of the system parameters, offering a clear overview and insights into their practical applications. Fig 12 provides an overview of the two models we used to compare the values between the RSSCA system and a conventional PV system.

Using the parameters in "Table 2", we simulated light distribution in the crop-growing area beneath the RSSCA system (a) and conventional PV system (b), with results shown in Fig 13. For an objective comparison of light distribution between the two systems, we analyzed data from a central area of 5000 mm × 5000 mm.

Based on simulated light distribution in the crop-growing area beneath the RSSCA system (a) and the conventional PV system (b), results indicate that the RSSCA system achieves a notably higher light uniformity, with 68% compared to 35% in the conventional PV system. In the conventional PV setup, light cannot pass through the solar panels themselves, so the crops primarily receive diffused light from the sky and limited light entering through panel gaps. Conversely, the RSSCA system allows diffuse light to pass freely through the Fresnel lenses, unobstructed, while also transmitting a portion of red light from direct sunlight through the RSSCA. This balanced light combination helps create a more consistent light environment across the crop-growing area, suggesting that plants under the RSSCA system may experience more uniform growth and potentially improved health compared to those grown under the conventional PV system.

To assess the system's efficiency on plants, the plants' light requirements need to be considered, typically defined by the Daily Light Integral (DLI) or the total Photosynthetically Active Radiation (PAR) received by the plant throughout the day [16], measured in moles of photons (mol$_{PAR}$). Plants' DLI needs are usually classified into four categories: low-light (< 10 mol$_{PAR}$/m$^2$/day), moderate-light (10–20 mol$_{PAR}$/m$^2$/day), high-light (20–30 mol$_{PAR}$/m$^2$/day), and very high-light (> 30

**Table 1. Specifications of prototypes of RSSCA system.**

| Parameter of RSSCA system | |
|---|---|
| Optical efficiency for direct sunlight [%] | 78 |
| Cell type | Triple-junction (GaInP/GaInAs/Ge) |
| Solar cell area [mm$^2$] | 20 mm × 20 mm |
| Lens type | PMMA Fresnel lens |
| Lens size | 250 mm × 250 mm |
| Geometrical concentration ratio, Cg [×] (= unit lens aperture area/unit cell active area) | 156 |
| Sun-tracking | 2-axis |

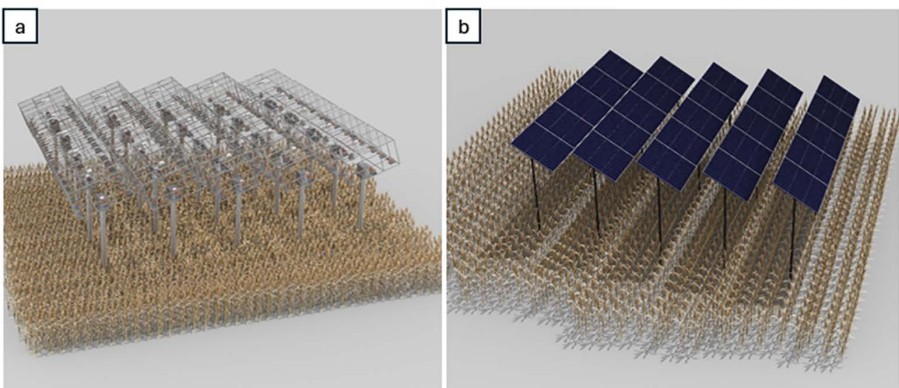

**Fig 12. Structural comparison of agrivoltaic models – RSSCA system vs. conventional PV system (a).** RSSCA; and (b) Traditional PV. The RSSCA system utilizes dual-axis tracking and modular Fresnel lenses for efficient light splitting and crop illumination, while the conventional PV model blocks light with static, opaque panels, reducing the quality of light available for plant growth.

**Table 2. Technical specifications of RSSCA system and conventional PV Systems.**

| Parameter | RSSCA system | Conventional PV system |
|---|---|---|
| Total area | 64.0 m² of Fresnel lenses on 100.0m² area | 64.0 m² of PV panel on 100.0 m² area |
| System height | 3.0 m | 3.0 m |
| Distance between modules | 1.5 m | 1.5 m |
| Tilt angle | Using sun-tracking | 30º towards South |
| Type solar panel | Multi-junction solar cells | Monocrystalline solar cells |
| Cells size | 20×20 mm | 1000×2000 mm |
| Solar-cell efficiency | 40% | 20% |

$mol_{PAR}$/m²/day) [24]. PAR, which encompasses the spectral range of solar radiation from 400 nm to 700 nm, is used by plants for photosynthesis. It is measured as Photosynthetic Photon Flux Density (PPFD), with units in $mol_{PAR}$/m²/s, where 1 $mol_{PAR}$/m²/s equals 6.022 x $10^{23}$ photons striking a m² area per second. To convert PAR from the solar spectrum in W/m², the photon flux spectrum in the 400–700 nm range (photons.m$^{-2}$.s$^{-1}$) is integrated over the entire spectrum radiation (W.m$^{-2}$) [21].

$$C_{PAR} = \eta_0 \frac{\int_{400}^{700} \varphi(\lambda)d\lambda}{\int_0^\infty I(\lambda)d\lambda}$$

(3)

In equation (3), $\eta_0$ represents the transmission efficiency of diffused sunlight; $\varphi(\lambda)$ denotes the incident photon flux spectrum; and $I(\lambda)$ is the spectrum measured in W/m². Since the proposed system is designed to utilize specific wavelengths of direct solar radiation, namely 420–470 nm and 600–655 nm for plant growth, the photosynthetically active radiation (PAR) from direct irradiation is calculated using the following formula [21]:

$$C_{PAR} = \eta_1 \frac{\int_{420}^{470} \varphi(\lambda)d\lambda}{\int_0^\infty I(\lambda)d\lambda} + \eta_2 \frac{\int_{600}^{655} \varphi(\lambda)d\lambda}{\int_0^\infty I(\lambda)d\lambda}$$

(4)

where the transmission efficiencies of the filter for the blue and red regions are represented by $\eta_1$ and $\eta_2$, respectively.

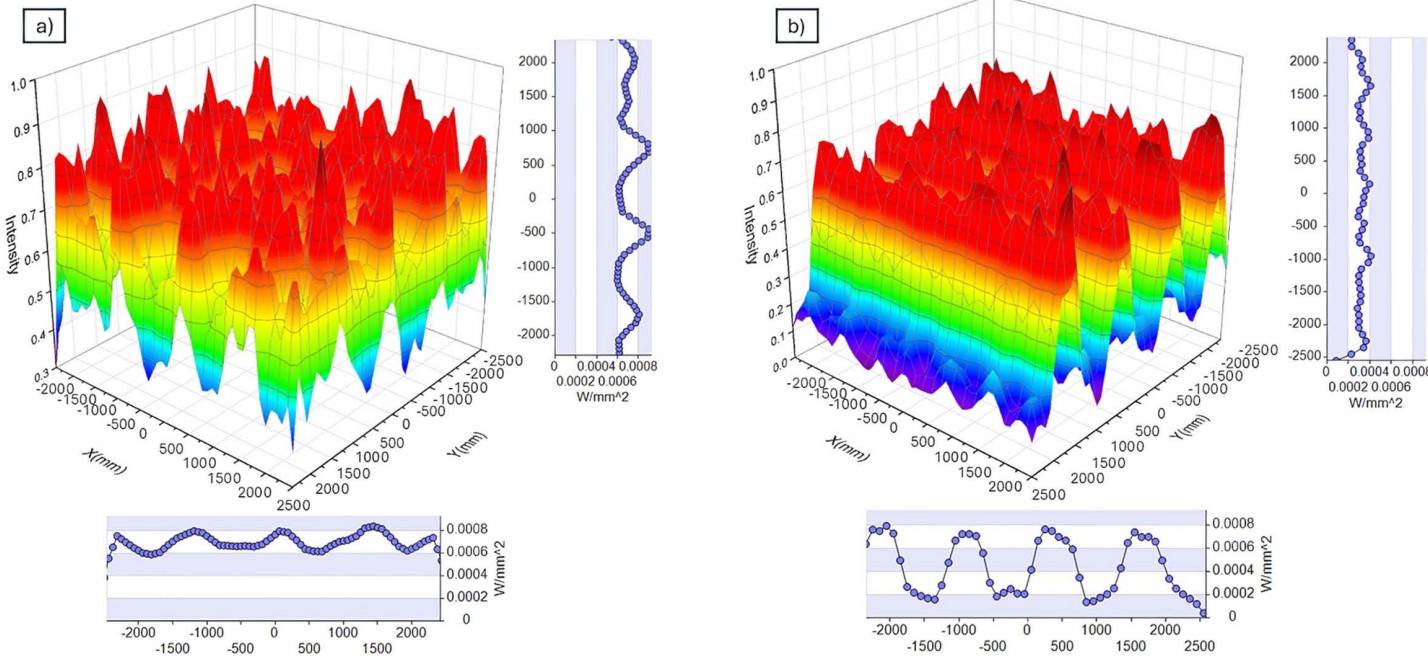

**Fig 13. Simulated ground-level light distribution – enhanced uniformity under RSSCA design (a) In crop-growing area under RSSCA systems, and (b) Conventional PV system.** Light distribution beneath the RSSCA system shows 68% uniformity, compared to only 35% under conventional PV, promoting balanced crop development through optimized diffuse and spectrally targeted light delivery.

To calculate $C_{PAR\ Direct}$ and $C_{PAR\ Diffuse}$ for each system, we performed simulations under three different conditions: bare land, CPV system, and conventional PV system. For bare land without any solar panel system and conventional PV system, we used the reference spectrum AM1.5 from NREL and applied formula 3 for diffuse light and direct light to calculate the PAR of the system for both light components. The results obtained are: $C_{PAR\ Direct} = 1.8\ \mu mol_{PAR}\ s^{-1}/W_{Solar}$ and $C_{PAR\ Diffuse} = 2.42\ \mu mol_{PAR}\ s^{-1}/W_{Solar}$. In the case of the PV system: $C_{PAR\ Direct} = 0.54\ \mu mol_{PAR}\ s^{-1}/W_{Solar}$ and $C_{PAR\ Diffuse} = 0.96\ \mu mol$ PAR $s^{-1}/W_{Solar}$. For the CPV system, we performed simulations to evaluate the energy and light spectrum received on the ground from both diffuse and direct light. Using formula 3 for diffuse light and formula 4 for direct light, the results are: $C_{PAR\ Direct} = 0.21\ \mu mol_{PAR}\ s^{-1}/W_{Solar}$ and $C_{PAR\ Diffuse} = 2.2\ \mu mol_{PAR}\ s^{-1}/W_{Solar}$. The detailed results are presented in Table 2.

The significant difference between $C_{PAR\ Direct}$ and $C_{PAR\ Diffuse}$ values in these cases can be explained as follows: In the CPV system, most of the radiation is concentrated on the solar cells to generate electricity, only a small fraction of the light in the red wavelength region is transmitted through the filter, and direct light from the space between the modules is considered. In addition, the transmission efficiency of Fresnel lens sheets, with a level of 93%, is also considered in determining $C_{PARDiffuse}$. This ensures that some light still reaches the ground, supporting plant growth. In the case of the conventional PV system, a large part of the agricultural land under the panels is affected by shading from the solar panel system, significantly reducing the amount of direct and diffuse light reaching the plants. This leads to lower $C_{PAR\ Direct}$ and $C_{PAR\ Diffuse}$ values in the conventional PV system. The arrangement of PV rows parallel and tilted 30 degrees to the south, although optimizing solar energy collection for the panels, creates large shadows, reducing the amount of light that can reach the plants below.

A detailed comparison of these systems in Table 3 helps us to better understand the impact of each system on plants growth and photosynthetic efficiency under real-world conditions. This result provides an important basis for developing energy-efficient and environmentally friendly agricultural renewable energy solutions.

 

**Table 3. Detailed Comparison of $C_{PAR\ Direct}$ and $C_{PAR\ Diffuse}$ for RSSCA and conventional PV Systems.**

|  | RSSCA System | Conventional PV System |
|---|---|---|
| Occupancy | 64% | 64% |
| $C_{PAR\ Direct}$ (µmol PAR s$^{-1}$/W$_{Solar}$) | 0.21 | 0.54 |
| $C_{PAR\ Diffuse}$ (µmol PAR s$^{-1}$/W$_{Solar}$) | 2.2 | 0.96 |

In this simulation, we also conducted an evaluation of the power generation efficiency of both systems. The results show that the power generation efficiency of the RSSCA system is 31.2%, which is significantly higher than that of the conventional PV system, which is only 20%. The higher efficiency of the RSSCA system can be explained by the concentration of sunlight on the solar cells, optimizing the conversion of light energy into electrical energy.

Based on the $C_{PAR\ Direct}$ and $C_{PAR\ Diffuse}$ values calculated in the previous section, we proceeded to calculate the DLI for three locations representing distinct climatic conditions: Hanoi (Vietnam), Ho Chi Minh City (Vietnam), and Seoul (Korea). Ho Chi Minh City, located near the equator, enjoys year-round sunlight with minimal seasonal variation. Hanoi, with its four distinct seasons, has a high annual solar radiation total, and even in winter, the levels of direct and diffuse radiation remain relatively high and comparable. In contrast, Seoul experiences a long winter with generally low radiation levels, where diffuse radiation surpasses direct radiation. These calculations were conducted for four representative days, capturing scenarios with varying light conditions: similar direct and diffuse radiation intensities, higher direct radiation than diffuse radiation, and lower direct radiation than diffuse radiation.

These calculations aim to provide a comprehensive and detailed picture of how the RSSCA and conventional PV systems will perform under different lighting conditions at the three study locations. Hanoi and Ho Chi Minh City are two major cities in Vietnam with different climate and solar radiation conditions, while Seoul represents a location in Korea with distinctly different climate conditions. The selected days allow us to compare and evaluate the performance of solar energy systems under diverse direct and diffuse radiation conditions (Table 4).

Based on the calculated DLI results, we observed that the DLI values for the RSSCA system in Hanoi and Ho Chi Minh City both exceed 14 mol/m²/day. This DLI level is sufficient to support rice growth, with the required DLI range for rice being between 15 and 25 mol/m²/day. This indicates that the RSSCA system in these regions has the potential to provide adequate light to support photosynthesis and rice growth, even under diffuse light conditions.

However, in Seoul, on days with lower direct light radiation intensity compared to diffuse light, the DLI values are very low, insufficient for rice growth. Specifically, these days, the RSSCA system cannot provide enough photosynthetic light

**Table 4. DLI calculations for different locations and electricity production capacity.**

|  |  | 20th Feb. | 30th May | 13th Aug. | 21st Nov. |
|---|---|---|---|---|---|
| DLI in Seoul (mol/m²/day) | RSSCA system | **20.3** | **38.5** | **26.2** | **10.2** |
|  | Conventional PV system | 18.8 | 21.7 | 12.1 | 4.5 |
| DLI in Hanoi (mol/m²/day) | RSSCA system | **14.3** | **28.7** | **32.2** | **22.3** |
|  | Conventional PV system | 6.6 | 15.3 | 16.1 | 13.4 |
| DLI in Ho Chi Minh City (mol/m²/day) | RSSCA system | **30.3** | **37.8** | **28.8** | **24.3** |
|  | Conventional PV system | 18.2 | 14.4 | 17.9 | 14.5 |
| Power Generation in Seoul (W/m²/day) | RSSCA system | **1520** | **2860** | **2014** | **73** |
|  | Conventional PV system | 420 | 510 | 460 | **92** |
| Power Generation in Hanoi (W/m²/day) | RSSCA system | **1240** | **2290** | **2577** | **2102** |
|  | Conventional PV system | 250 | 430 | 580 | 421 |
| Power Generation in Ho Chi Minh City (W/m²/day) | RSSCA system | **1994** | **2790** | **2368** | **2330** |
|  | Conventional PV system | 708 | 480 | 563 | 520 |

to reach the minimum DLI required for rice. This highlights the RSSCA system's strong dependence on direct light conditions, and when this condition is insufficient, the system cannot compensate with diffuse light to maintain the necessary DLI level.

This analysis suggests that while the RSSCA system can operate effectively in regions like Hanoi and Ho Chi Minh City with sufficiently high DLI levels to support crops, in areas with lower direct light intensity like Seoul at certain times, this system may not meet the light requirements of rice plants. Therefore, the selection and design of solar energy systems need to carefully consider the solar radiation conditions of each region to ensure maximum efficiency and optimal support for agriculture.

The seasonal variations in electricity production capacity between the RSSCA and PV systems highlight the influence of direct versus diffuse light, as well as weather patterns. In the simulation results presented for Seoul, both RSSCA and conventional PV systems perform differently depending on the season, primarily due to the varying intensity and distribution of sunlight. During the summer months, such as August 13, the RSSCA system performs significantly better, generating 2014 W/m$^2$/day compared to the PV system's 460 W/m$^2$/day. This is because RSSCA systems rely heavily on direct sunlight, which is more abundant in summer due to clearer skies and a higher solar angle. The direct light enables the RSSCA system to capture and concentrate more energy, resulting in a 4.4 times higher output than the PV system. Similarly, on particularly sunny days with high levels of direct radiation, the RSSCA system outperforms the conventional PV system by an even greater margin, as shown by the results on days with intense sunlight, where RSSCA produces 2860 W/m$^2$/day, 5.6 times more than the PV system's 510 W/m$^2$/day. However, as we move into winter, such as November 21, the situation reverses. The RSSCA system generates only 73 W/m$^2$/day, while the PV system produces 92 W/m$^2$/day. This shift can be attributed to the lower intensity of direct sunlight during the winter months, combined with increased diffuse light due to cloud cover and the sun's lower position in the sky. Traditional PV systems can utilize both direct and diffuse light, giving them a slight advantage in low-light conditions, whereas RSSCA systems are less efficient when direct sunlight is limited.

For power generation in Hanoi and Ho Chi Minh City, the data clearly demonstrates the significant superiority of the RSSCA system compared to the conventional PV system. In Hanoi, where there are distinct seasonal variations and substantial diffuse radiation, the RSSCA system generates 4–6 times more power, showcasing its ability to efficiently harness diverse light conditions. Meanwhile, in Ho Chi Minh City, with stable sunlight throughout the year, the RSSCA system still outperforms, producing 3–5 times more power, though the performance gap is narrower due to the abundance of direct radiation. Overall, the RSSCA system exhibits exceptional versatility and efficiency, making it particularly suitable for regions with varying or diffuse solar radiation.

These differences underscore the importance of seasonality when evaluating the performance of CPV versus PV systems. In regions or times of the year with strong direct sunlight, CPV systems can deliver significantly higher energy yields. In contrast, during overcast or winter conditions, PV systems may offer better consistency due to their ability to capture diffuse sunlight. This suggests that CPV systems are best suited for locations or seasons with high direct radiation, while PV systems can maintain a more stable output year-round.

The results discussed in this section experimentally demonstrate the RSSCA system's red spectral splitting capability, which fully and effectively fulfills the light requirements for plant photosynthesis. In near future studies, a field trial setup will be developed to further validate the present research findings. Nevertheless, we anticipate that the experimental results will closely align with the current outcomes, as the input data conditions and the proposed system model closely replicate real-world scenarios.

## Conclusion

The Red Spectrum Splitting Concentrated Agrivoltaic (RSSCA) system introduced in this study addresses the pressing challenge of balancing agricultural productivity with the expansion of solar energy infrastructure. By selectively

transmitting red light (~630 nm)—crucial for rice photosynthesis—to crops and redirecting the remaining spectrum to high-efficiency multi-junction solar cells, the RSSCA system enables simultaneous crop growth and electricity generation without competition for land use.

Simulation and analytical modeling results show that the RSSCA system achieves a photoelectric conversion efficiency of 31.2%, significantly outperforming conventional PV systems, which typically achieve only around 20%. Daily Light Integral (DLI) values under RSSCA in Hanoi ranged from 14.3 to 32.2 mol/m²/day, and in Ho Chi Minh City from 24.3 to 37.8 mol/m²/day, both of which meet or exceed optimal requirements for rice cultivation. In contrast, conventional PV systems in the same regions only provide 6.6–16.1 mol/m²/day and 14.4–18.2 mol/m²/day, respectively.

In terms of electricity production, the RSSCA system yields up to 2790 W/m²/day in Ho Chi Minh City and 2577 W/m²/day in Hanoi—3–5 times higher than conventional systems under the same conditions. These quantitative findings confirm the RSSCA's superior dual-function capability and land-use efficiency, particularly for tropical and subtropical climates with high solar radiation.

However, performance declines in areas with low direct sunlight, such as Seoul in winter, where the system's reliance on direct irradiance limits its effectiveness. This suggests the potential for future improvements, such as incorporating diffuse light collection or hybrid solutions to maintain performance under varying weather conditions.

Overall, the RSSCA system offers a promising strategy for achieving food–energy co-optimization. Its ability to significantly enhance electricity generation while preserving, or even supporting, agricultural productivity makes it well-suited for sustainable development goals in land-scarce and sun-rich regions. Future research should focus on long-term field testing, crop-specific spectral tuning, and economic feasibility studies to accelerate real-world adoption and scalability.

## Author contributions

**Conceptualization:** Tran Quoc Tien, Ngoc-Hai Vu, Seoyong Shin.

**Data curation:** Ngoc-Minh Kieu, Thanh-Phuong Nguyen, Quang Cong Tong.

**Formal analysis:** Tran Quoc Tien, Ngoc-Hai Vu, Thanh-Phuong Nguyen.

**Investigation:** Ngoc-Minh Kieu, Quang Cong Tong.

**Writing – original draft:** Ngoc-Minh Kieu.

**Writing – review & editing:** Tran Quoc Tien, Ngoc-Hai Vu.

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
