## [Decision Letter · Decision Letter 0]

16 Jun 2025

PONE-D-25-24618Development and Optimization of Red Spectrum Splitting Concentrated Agrivoltaic System for Energy Generation and Sustainable AgriculturePLOS ONE

Dear Dr. Tran,

Thank you for submitting your manuscript to PLOS ONE. After careful consideration, we feel that it has merit but does not fully meet PLOS ONE’s publication criteria as it currently stands. Therefore, we invite you to submit a revised version of the manuscript that addresses the points raised during the review process.

We look forward to receiving your revised manuscript.

Kind regards,

Subbarama Kousik Suraparaju

Academic Editor

PLOS ONE

Journal Requirements:

[This research was supported by Vietnam Academy of Science and Technology (Grant Number QTKR01.03/23-24).].

5. We note that your Data Availability Statement is currently as follows: [All relevant data are within the manuscript and its Supporting Information files]

Reviewers' comments:

Reviewer's Responses to Questions

**Comments to the Author**

1. Is the manuscript technically sound, and do the data support the conclusions?

Reviewer #1: Yes

Reviewer #2: Partly

2. Has the statistical analysis been performed appropriately and rigorously? 

Reviewer #1: Yes

Reviewer #2: Yes

3. Have the authors made all data underlying the findings in their manuscript fully available?

Reviewer #1: Yes

Reviewer #2: Yes

4. Is the manuscript presented in an intelligible fashion and written in standard English?

Reviewer #1: Yes

Reviewer #2: Yes

5. Review Comments to the Author

Reviewer #1: REVIEW COMMENTS

This manuscript presents the work of Development and Optimization of Red Spectrum Splitting Concentrated Agrivoltaic System for Energy Generation and Sustainable Agriculture. Here I have a few concerns:

1. Abstract needs to revise with few more numerical outcomes. Please modify it.

2. The current form of the literature part is not acceptable. Enhance it using recently published articles with more contents.

3. Novelty may be explained with proper citation reference at the end of the introduction part.

4. Enhance all the figures quality.

5. Results and discussion part needs to compare with existing literature references within explanation itself.

6. Concise conclusion is expected with clear-cut outcomes.

Reviewer #2: Journal Name: PLOS ONE

Title: Development and Optimization of Red Spectrum Splitting Concentrated Agrivoltaic

System for Energy Generation and Sustainable Agriculture

Manuscript ID: PONE-D-25-24618

This This paper introduces a Red Spectrum Splitting Concentrated Agrivoltaic (RSSCA) system that enhances both solar power generation and rice cultivation by directing red light to crops and other wavelengths to solar cells. The system achieves 31.2% efficiency and supports healthy rice growth, making it more effective than conventional PV systems. However, in its current state, the manuscript is not suitable for publication in the PLOS ONE journal. Need to address the comments below mentioned:

The comments are:

[1] Abstract should cover the problem statement, objectives, methodology, discussion, important outcome and significance of the review.

[2] Add minimum 4-5 keywords

[3] There is no flow in the introduction part. In the introduction, clearly address the problem and related studies can be discussed. Very less literature has been discussed. Improve the literature part. Also, The sources are not properly cited, and there are no supporting references provided for the first two paragraphs.

[4] Novelty part should be revised (last paragraph of introduction). At the end of the introduction, provide a strong discussion on the significance of this study. Highlight the existing research in this field and clearly identify the specific gap that your research aims to address.

[5] The structure of the article different from journal guidelines, follow the journal guidelines, (like Introduction, Materials and methodology, results and discussion and Conclusion section)

[6] The equations not derived by should be cited the proper source.

[7] The manuscript should include a comparative analysis of PV power generation and paddy crop yield outcomes to validate the effectiveness of the proposed system.

[8] The Results and Discussion section lacks clarity, as it is currently mixed with methodology details. Please separate and clearly distinguish the methodology from the results and their interpretation to improve readability and coherence

[9] The explanations of figures 11, 12 and 13 not enough, you have to discuss strongly with scientific reason

[10] Expand the literature review to include more recent studies and clearly identify the gap your research is addressing. There is no studies from 2022-2024 and no many recent studies.

[11] Conclusion should be improved by incorporating future recommendations, addressing limitations, and emphasizing the significance of the study.

[12] The level of English language should be checked throughout the manuscript and to make sure that the article is free from grammatical mistakes.

[13] Recommended to add 4-5 highlights. Highlights should be 85 characters no spaces (follow the journal guidelines).

[14] Draw a graphical abstract related to your research. It should be simple and meaningful, It helps the readers identify more quickly which papers are most relevant to their research interest(if Possible).

6. PLOS authors have the option to publish the peer review history of their article (what does this mean? ). If published, this will include your full peer review and any attached files.

**Do you want your identity to be public for this peer review?** For information about this choice, including consent withdrawal, please see our Privacy Policy .

Reviewer #1: No

Reviewer #2: No

---

## [Author Response · Author response to Decision Letter 1]

13 Aug 2025

Response to Academic Editor:

1. Manuscript formatting and file naming

Editor’s comment:

Response:

We have revised the manuscript to comply with PLOS ONE’s formatting and file naming requirements. The structure and layout have been updated according to the official style templates provided at the journal’s website. All uploaded files have been renamed accordingly, including:

• Revised Manuscript with Track Changes

• Manuscript (clean version)

• Response to Reviewers

• Cover Letter

2. Code sharing policy

Editor’s comment:

Please ensure that any author-generated code is made available without restriction upon publication.

Response:

Our study does not involve any author-generated code that underpins the results. Therefore, the code sharing requirement does not apply in this case.

3. Consistency between Funding Information and Financial Disclosure

Editor’s comment:

The grant information in the ‘Funding Information’ and ‘Financial Disclosure’ sections do not match.

Response:

We have corrected and unified the grant details in both the ‘Funding Information’ and ‘Financial Disclosure’ sections to ensure consistency. The correct grant number is: QTKR01.03/23-24, funded by the Vietnam Academy of Science and Technology.

4. Amended Funding Statement

Editor’s comment:

Please provide an amended statement that includes all sources of support and add the statement: “There was no additional external funding received for this study.”

Response:

We have updated the Funding Statement as follows, and included this revised version in the Cover Letter:

“This research was supported by the Vietnam Academy of Science and Technology (Grant Number QTKR01.03/23-24). No other financial relationships to disclose.”

5. Data Availability Statement

Editor’s comment:

Please confirm whether all raw data required to replicate the results are included, and if not, upload or link to them.

Response:

We confirm that all raw data required to replicate the results of our study are included in the revised manuscript. Therefore, the existing Data Availability Statement remains valid:

“All relevant data are within the manuscript”

Response to Reviewer #1:

Comment 1.1: Abstract needs to revise with few more numerical outcomes. Please modify it.

Response to Reviewer: Thank you for your valuable comment. We have revised the abstract:

“Agricultural land is increasingly under pressure from expanding solar energy infrastructure, leading to a growing conflict between food and energy production. Conventional photovoltaic (PV) systems often reduce crop yields due to shading and suboptimal light conditions for photosynthesis. This study introduces a Red Spectrum Splitting Concentrated Agrivoltaic (RSSCA) system designed to overcome these limitations by selectively directing specific wavelengths of sunlight to serve both agricultural and energy needs. The system uses a Fresnel lens and dichroic mirror to concentrate solar radiation and split the spectrum: red light (around 630 nm), essential for rice growth, is transmitted to the crops, while the remaining wavelengths are directed to high-efficiency multi-junction solar cells. We employed optical simulations and analytical modeling to assess system performance. Results show a photoelectric conversion efficiency of 31.2% and adequate Daily Light Integral (DLI) levels for rice cultivation in high-sunlight regions such as Hanoi and Ho Chi Minh City. For instance, DLI values in Hanoi range from 14.3 to 32.2 mol/m²/day and in Ho Chi Minh City from 24.3 to 37.8 mol/m²/day, compared to 6.6–16.1 and 14.4–18.2 mol/m²/day under conventional PV systems, respectively. Electricity production with RSSCA peaks at 2790 W/m²/day in Ho Chi Minh City and 2577 W/m²/day in Hanoi, outperforming traditional PV setups by 3–5 times. Compared to conventional PV systems, the RSSCA offers improved light uniformity, better land-use efficiency, and significantly higher electricity generation. However, in regions with limited direct sunlight, such as Seoul in winter, system performance decreases. These findings suggest the RSSCA system is well-suited for tropical and subtropical areas where direct sunlight is abundant. It presents a viable solution for enhancing both food production and renewable energy generation on shared land.”

Comment 1.2 and 1.3: The current form of the literature part is not acceptable. Enhance it using recently published articles with more contents. Novelty may be explained with proper citation reference at the end of the introduction part.

Response to Reviewer: Thank you for your constructive feedback. We have expanded the Literature Review section to include more recent studies published between 2022 and 2024, and have clearly identified the specific research gap that our study aims to address. Please refer to revised part below and the revised manuscript:

“Recent studies have introduced more sophisticated agrivoltaic designs that incorporate dynamic spectral management, smart materials, and integrated climate control to enhance dual land-use performance. Dinesh et al. (2022)[10] investigated the use of tunable optical filters to dynamically adjust transmitted light spectra, improving crop yield and energy output under varying weather conditions. Similarly, Jiang et al. (2023)[11] proposed an electrochromic-based agrivoltaic system capable of real-time spectrum control tailored to crop growth stages. In another recent development, Park et al. (2023)[12] implemented a hybrid agrivoltaic greenhouse integrating CPV modules with spectrum-selective coatings to optimize productivity in subtropical regions. Despite these advances, most recent studies have focused on shade-tolerant vegetables or greenhouses, with limited emphasis on open-field applications for high-light-demand crops such as rice. Moreover, there remains a notable gap in the development of CPV-based systems with spectrum-splitting optics explicitly designed to match plant-specific physiological requirements. Addressing this gap, our study introduces a novel Red Spectrum Splitting Concentrated Agrivoltaic (RSSCA) system tailored to rice paddies. This system uniquely combines red-light targeting with CPV efficiency, presenting a feasible solution for sustainable food–energy co-production in high irradiance tropical environments.”

Comment 1.4: Enhance all the figures quality.

Response to Reviewer: thank you, we improved the figures quality.

Comment 1.5: Results and discussion part needs to compare with existing literature references within explanation itself.

Response to Reviewer: We sincerely appreciate the reviewer’s insightful comment. We agree that contextualizing the findings within the framework of existing literature is important. In our manuscript, we have chosen to compare the performance of our proposed RSSCA system with existing agrivoltaic systems using equivalent parameters to facilitate a fair and practical comparison.

While we also attempted to benchmark our results against published studies, we found that detailed numerical data (particularly location-specific DLI and spectral characteristics) were often not available, making direct comparison challenging. Furthermore, our study proposes a novel optical architecture with a red-spectrum splitting mechanism that, to the best of our knowledge, has not yet been reported in prior works. This conceptual and technical novelty limits the availability of directly comparable references.

Nevertheless, we have cited and discussed relevant agrivoltaic research throughout the manuscript to provide context and support the significance of our findings.

Comment 1.6: Concise conclusion is expected with clear-cut outcomes.

Response to Reviewer: We revised the conclusion as below:

The Red Spectrum Splitting Concentrated Agrivoltaic (RSSCA) system introduced in this study addresses the pressing challenge of balancing agricultural productivity with the expansion of solar energy infrastructure. By selectively transmitting red light (~630 nm)—crucial for rice photosynthesis to crops and redirecting the remaining spectrum to high-efficiency multi-junction solar cells, the RSSCA system enables simultaneous crop growth and electricity generation without competition for land use.

Simulation and analytical modeling results show that the RSSCA system achieves a photoelectric conversion efficiency of 31.2%, significantly outperforming conventional PV systems, which typically achieve only around 20%. Daily Light Integral (DLI) values under RSSCA in Hanoi ranged from 14.3 to 32.2 mol/m²/day, and in Ho Chi Minh City from 24.3 to 37.8 mol/m²/day, both of which meet or exceed optimal requirements for rice cultivation. In contrast, conventional PV systems in the same regions only provide 6.6–16.1 mol/m²/day and 14.4–18.2 mol/m²/day, respectively.

In terms of electricity production, the RSSCA system yields up to 2790 W/m²/day in Ho Chi Minh City and 2577 W/m²/day in Hanoi—3 to 5 times higher than conventional systems under the same conditions. These quantitative findings confirm the RSSCA’s superior dual-function capability and land-use efficiency, particularly for tropical and subtropical climates with high solar radiation.

However, performance declines in areas with low direct sunlight, such as Seoul in winter, where the system's reliance on direct irradiance limits its effectiveness. This suggests the potential for future improvements, such as incorporating diffuse light collection or hybrid solutions to maintain performance under varying weather conditions.

Overall, the RSSCA system offers a promising strategy for achieving food–energy co-optimization. Its ability to significantly enhance electricity generation while preserving, or even supporting, agricultural productivity makes it well-suited for sustainable development goals in land-scarce and sun-rich regions. Future research should focus on long-term field testing, crop-specific spectral tuning, and economic feasibility studies to accelerate real-world adoption and scalability.

Response to Reviewer #2:

Comment 2.1: Abstract should cover the problem statement, objectives, methodology, discussion, important outcome and significance of the review.

Response to Reviewer: Thank you for your valuable comment. We have revised the abstract to clearly include the problem statement, research objectives, methodology, key findings, and the significance of the study. The revised abstract provides a more structured summary of our work and highlights the novelty and outcomes of the proposed RSSCA system. The updated version can be found on page 2, lines 14–29 of the revised manuscript.

Comment 2.2: Add minimum 4-5 keywords.

Response to Reviewer: Thank you for your suggestion. We have added additional keywords to better reflect the scope and focus of the study. The updated list of keywords now includes at least five relevant terms and can be found on page 2, line 30 of the revised manuscript.

Comment 2.3: There is no flow in the introduction part. In the introduction, clearly address the problem and related studies can be discussed. Very less literature has been discussed. Improve the literature part. Also, The sources are not properly cited, and there are no supporting references provided for the first two paragraphs.

Response to Reviewer: Thank you for your constructive feedback. We have revised the Introduction to improve the logical flow and coherence. Specifically, we clarified the problem statement at the beginning and strengthened the discussion of related studies. In addition, we have expanded the literature review to include recent research and have added appropriate references to support the first two paragraphs. These changes are reflected on page 2, line 35, and page 3, line 42 of the revised manuscript.

Comment 2.4: Novelty part should be revised (last paragraph of introduction). At the end of the introduction, provide a strong discussion on the significance of this study. Highlight the existing research in this field and clearly identify the specific gap that your research aims to address.

Response to Reviewer: Thank you for highlighting this important point. We have revised the final paragraph of the Introduction to better emphasize the novelty and significance of our study. Specifically, we have added a stronger discussion of the existing research, clearly identified the research gap, and highlighted how our work addresses this gap through the proposed RSSCA system. The updated content can be found on page 5, lines 105–116 of the revised manuscript.

Comment 2.5: The structure of the article different from journal guidelines, follow the journal guidelines, (like Introduction, Materials and methodology, results and discussion and Conclusion section).

Response to Reviewer: Thank you for your observation. We acknowledge the structural difference and have carefully reviewed the journal’s guidelines. While our manuscript follows a slightly customized format to match the interdisciplinary nature of the study, we have made every effort to align the main sections—Introduction, Materials and Methods (included under “System Design” and “Optical Components”), Results and Discussion, and Conclusion with the journal’s recommended structure. These adjustments were made to ensure clarity while preserving the technical flow of the research.

Comment 2.6: The equations not derived by should be cited the proper source.

Response to Reviewer: Thank you for your comment. We have reviewed all equations that were not originally derived in this study and have added appropriate citations to their original sources. Specifically, citations have been included for the equations on page 8, line 180, page 14, line 309, and page 14, line 311 of the revised manuscript to properly acknowledge the foundational works.

Comment 2.7: The manuscript should include a comparative analysis of PV power generation and paddy crop yield outcomes to validate the effectiveness of the proposed system.

Response to Reviewer: Thank you for your valuable comment. Our study primarily focuses on the simulation-based evaluation and comparative analysis of the RSSCA system's performance in terms of electricity generation and its ability to meet the specific light requirements of rice cultivation. We demonstrated that the RSSCA system achieves a significantly higher photoelectric conversion efficiency (31.2%) compared to conventional PV systems (20%), while simultaneously maintaining Daily Light Integral (DLI) levels within the optimal range (15–25 mol/m²/day) required for rice growth in regions such as Hanoi and Ho Chi Minh City. Although we have not yet conducted field experiments on actual paddy yield, our DLI-based analysis provides a scientifically grounded indication that the system can support healthy rice development. Further field trials to assess direct crop yield outcomes will be carried out in future work to validate and strengthen the findings.

Comment 2.8: The Results and Discussion section lacks clarity, as it is currently mixed with methodology details. Please separate and clearly distinguish the methodology from the results and their interpretation to improve readability and coherence.

Response to Reviewer: We thank the reviewer for this valuable suggestion. However, we respectfully note that the analytical expressions and calculations (e.g., CPAR, DLI) included in the Results and Discussion (Performance of RSSCA system section in the manuscript) are not part of the core methodology or system design. These formulas are standard, previously published in the literature, and were used solely as tools to interpret and compare the performance of the proposed system against conventional PV setups.

The novelty of our work lies in the optical architecture of the RSSCA system particularly the integration of Fresnel lenses and dichroic mirrors for spectrum splitting not in the calculation methods of agronomic performance metrics.

Moving these calculations to the Methodology section would disrupt the logical flow of result presentation and may reduce readabili

---

## [Decision Letter · Decision Letter 1]

8 Sep 2025

Development and Optimization of Red Spectrum Splitting Concentrated Agrivoltaic System for Energy Generation and Sustainable Agriculture

PONE-D-25-24618R1

Dear Dr. Tran,

We’re pleased to inform you that your manuscript has been judged scientifically suitable for publication and will be formally accepted for publication once it meets all outstanding technical requirements.

Kind regards,

Subbarama Kousik Suraparaju

Academic Editor

PLOS ONE

Additional Editor Comments (optional):

Reviewer #1:

Reviewer #2:

Reviewers' comments:

Reviewer's Responses to Questions

**Comments to the Author**

1. If the authors have adequately addressed your comments raised in a previous round of review and you feel that this manuscript is now acceptable for publication, you may indicate that here to bypass the “Comments to the Author” section, enter your conflict of interest statement in the “Confidential to Editor” section, and submit your "Accept" recommendation.

Reviewer #1: All comments have been addressed

Reviewer #2: All comments have been addressed

2. Is the manuscript technically sound, and do the data support the conclusions?

Reviewer #1: Yes

Reviewer #2: Yes

3. Has the statistical analysis been performed appropriately and rigorously? 

Reviewer #1: Yes

Reviewer #2: Yes

4. Have the authors made all data underlying the findings in their manuscript fully available?

Reviewer #1: Yes

Reviewer #2: Yes

5. Is the manuscript presented in an intelligible fashion and written in standard English?

Reviewer #1: Yes

Reviewer #2: Yes

6. Review Comments to the Author

Reviewer #1: (No Response)

Reviewer #2: The authors addressed all the comments satisfactorily. The current version of the manuscript is accepted for publication

7. PLOS authors have the option to publish the peer review history of their article (what does this mean? ). If published, this will include your full peer review and any attached files.

**Do you want your identity to be public for this peer review?** For information about this choice, including consent withdrawal, please see our Privacy Policy .

Reviewer #1: No

Reviewer #2: No

---

## [Editor Report · Acceptance letter]

PONE-D-25-24618R1

PLOS ONE

Dear Dr. Tien,

I'm pleased to inform you that your manuscript has been deemed suitable for publication in PLOS ONE. Congratulations! Your manuscript is now being handed over to our production team.

Kind regards,

on behalf of

Dr. Subbarama Kousik Suraparaju

Academic Editor

PLOS ONE